# Incidence and preventive treatment for deep vein thrombosis with our own preventive protocol in total hip and knee arthroplasty

Takashige Momose[1], Masaki Nakano[2], Yukio Nakamura [2]*, Takashi Maeda[1], Masashi Nawata[1]

1 Department of Orthopaedic Surgery, Marunouchi Hospital, Matsumoto, Nagano, Japan, 2 Department of Orthopaedic Surgery, Iida Hospital, Iida, Nagano, Japan

☯ These authors contributed equally to this work.
* yxn14@aol.jp

**Data Availability Statement:** All relevant data are within the paper and its Supporting Information files.

## Abstract

The aim of the present study was to investigate the perioperative and postoperative incidence of deep vein thrombosis (DVT) and validate the effectiveness of our own preventive treatment protocol for venous thromboembolism (VTE) occurrence in lower extremity arthroplasty patients. The subjects were 1,054 patients (mean age: 74.3 years) who underwent total hip arthroplasty (THA) or total knee arthroplasty (TKA) at our institutions between April 2014 and March 2017. We examined the frequencies of pre- and post-operative DVT by lower extremity Doppler images, and the incidence rate at proximal or distal regions as well as that according to preoperative DVT status were evaluated. Preoperative DVT was detected in 6.5% (69 cases) of our cohort and those were located 1.4% (15 cases) at proximal and 5.1% (54 cases) at distal regions. A significantly higher rate of postoperative DVT development was observed in preoperative DVT+ THA patients ($P = 0.0075$), but not in TKA patients only with a higher tendency ($P = 0.56$). The overall incidence of DVT up to 2 weeks after surgeries was 27.3% (288 cases); however, the rate in proximal femur regions was suppressed to 2.8% (30 cases), and there was no symptomatic pulmonary thromboembolism (PTE) case. The results demonstrated the importance of regular Doppler examination for early detection of postoperative DVT occurrence and the following immediate treatment initiation. Our own VTE preventive treatment protocol could reduce the development of proximal DVT, and the periodic monitoring as well as prompt treatment might prevent the fatal PTE. osteoarthritis (OA), rheumatoid arthritis (RA).

## Introduction

Since venous thromboembolism (VTE) is a major cause of morbidity, mortality, and healthcare costs in arthroplasty patients, the prevention of VTE occurrence should be a universal quality improvement initiative. Lower extremity arthroplasty (total hip arthroplasty; THA and total knee arthroplasty; TKA) has been classified as a high risk procedure for VTE incidents

**Funding:** The authors received no specific funding for this work.

**Competing interests:** The authors have declared that no competing interests exist

[1], and medical history of VTE reportedly increases the risk further [2]; thus, preoperative assessment of deep vein thrombosis (DVT) in patients scheduled for THA and/or TKA will be of importance to postoperative VTE prevention and joint function recovery [3].

Approximately 1 case in 71 patients and 1 case in 167 patients undergoing TKA and THA procedures, respectively, reportedly developed a VTE within 30 days after surgery [3]. Recently, Sloan et al. have suggested that overweight or obesity was associated with an increased risk of pulmonary embolism (PE) occurrence but not DVT after primary THA or TKA by a large administrative database study [4]. Moreover, in Japanese population, the incidence of DVT, pulmonary thromboembolism (PTE), and bleeding events as well as the risk factors for DVT and PTE were demonstrated to be comparable with those in overseas, and such risk factors for DVT and/or PTE as female sex, VTE history, and thrombophilia were statistically significant [5].

The most common strategies for preventing the development of DVT and PTE in patients with a higher risk include adequate anticoagulant therapy and intermittent pneumatic compression in the early postoperative period. Although a number of strategies as well as guidelines have been implemented to reduce the VTE occurrence, their impacts remain unclear. Based on a research of literature published between 2004 and 2014 including 13 studies, Wilson et al. described that aspirin might be considered a suitable alternative to thromboprophylactic agents following THA and TKA [6].

However, there have still been few reports on the preventive treatment protocol before arthroplasty procedures, which is intended to reduce the proximal DVT and symptomatic PTE. The aim of the present study was to investigate the incidence of pre- and post-operative DVT and validate the effectiveness of our own preventive treatment protocols for VTE occurrence in THA and TKA patients perioperatively.

## Methods

The subjects of this investigation were 1,054 patients (202 males and 852 females) who underwent THA (501 cases, 587 joints) and TKA (553 cases, 900 joints) at our institutions between April 2014 and March 2017. Approximately one year between June in 2021 to May 2022 were accessed for research purposes. The average age at operation was 74.3 years (43–91 years), and 1003 cases were osteoarthritis (OA; hip 468 and knee 535), 41 cases were rheumatoid arthritis (RA; hip 25 and knee 16), 8 cases were osteonecrosis of the femoral head, and 2 cases were idiopathic osteonecrosis of the knee. We had access to information that could identify individual participants during and/or after data collection.

The study design in this study was as the following 1), and 2) a) to d).

1. Subjects were the patients with THA or TKA

2. Perioperative use of our own VTE preventive protocol;

    a. Evaluation of VTE in lower extremity using echography preoperatively (3 months and 1 month before) and postoperatively

    b. Preoperative or postoperative administration of NOAC(s) or Warfarin appropriately.

    c. Use of gradual compression stocking (GCS) and A venous foot pump (VF) (A-V Impulse™, Covidien, MN, USA)

    d. ROM exercize and walking practice with physical therapists after surgery

We firstly investigated the frequency of preoperative DVT by using lower extremity Doppler images. The location of DVT occurrence (proximal or distal) was examined, and such DVT

risk factors as age, sex, body mass index (BMI), plasma D-dimer, and preoperative gait function were compared between the patients with DVT (DVT+) and without DVT (DVT−). Gait functions were assessed by the Japanese Orthopaedic Association Score and the Function Score of Knee Society Score for THA and TKA patients, respectively. After surgeries, the incidence of postoperative DVT was investigated at the first and second weeks and assessed according to the presence or absence of preoperative DVT. The occurrence of PTE was also examined postoperatively. VF was applied from just after surgery to 3 days and GCS was used for 2 weeks after surgery. Mobilization was started on the next day of surgery, and then range of motion exercize and walking practice with physical therapists at the institutions were performed until discharge. Physiotherapy was not performed during this study.

In our own protocol for VTE prophylaxis (Fig 1), lower extremity Doppler echography for DVT screening was performed in all patients who had been planned to have surgeries, and an anticoagulant medication was undertaken in those patients with preoperative distal DVT. In those patients, surgeries were performed with novel oral anticoagulants (NOACs) medication. In the cases of large proximal thrombi, firstly, each patient consulted a certified specialist in the field, secondly, thoracic computed tomography (CT) and CT angiography were performed to ascertain the existence of PTE, and finally, whether or not having surgery was possible had been evaluated by the certified specialists and our surgery team. The patients with preoperative DVT had Doppler examination again just before surgery to validate the thrombotic status. Postoperatively, a physical prophylaxis was conducted immediately after the surgery, and NOACs were administered from the day after surgery for 1 week. Lower extremity Doppler imaging was carried out at week(s) 1 and 2 postoperatively for all patients, and treatment was started upon the occurrence of DVT (Fig 1).

We have routinely performed preoperative lower limb Doppler ultrasound examination in all patients scheduled for THA and TKA to screen DVT and practiced common perioperative prophylactic treatments, such as NOACs or warfarin, because there have been few data about when, how much, and how long those treatments should be used perioperatively in order to prevent fatal PTE. The basic ideas for our VTE prophylaxis protocol are as follows: 1) Preoperative screening of pre-existing DVT by lower extremity Doppler images to reduce the risk of postoperative VTE, 2) Prevention of massive PTE development using NOACs or warfarin from the day after surgery for 1 week, and 3) Periodical evaluation for DVT and immediate initiation of treatment upon the occurrence of DVT (Table 1).

In the present study, we firstly examined the prevalence of preoperative DVT in patients who were scheduled for THA and TKA procedures, and then compared the risk factor parameters according to the presence or absence of DVT preoperatively as well as the frequencies of postoperative DVT occurrence. Statistical analyses were performed by Student's *t*-test, Mann–Whitney U test, and Chi-squared testing. A two-tailed *P*-value of $< 0.05$ was considered significantly different. All statistical tests were carried out by R version 3.6.0 software (https://www.r-project.org/).

## Results

The overall frequency of preoperative DVT detection in our cohort was 6.5% (69 cases/1054 cases) (Table 2). Preoperative lower extremity Doppler imaging detected DVT in 39 cases (7.8%) of THA planned patients; of these, proximal regions were 9 cases and distal regions were 30 cases. Similarly, as for TKA scheduled patients, DVT was preoperatively observed in 30 cases (5.4%) which were located 6 cases at proximal regions and 24 cases at distal regions (Table 2).

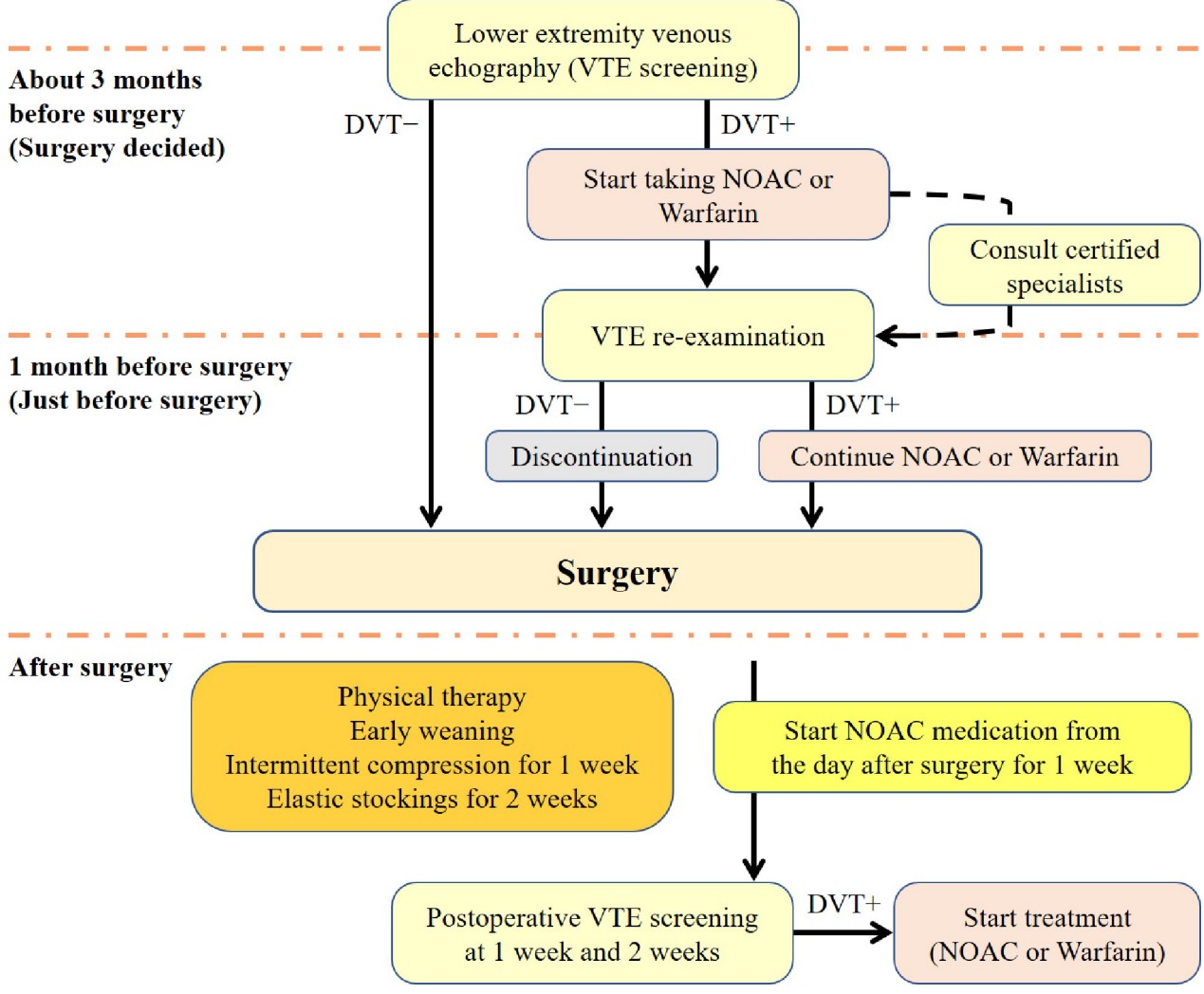

**Fig 1. Our own preventive treatment protocol for VTE.** VTE, venous thromboembolism; DVT, deep vein thrombosis; NOAC, novel oral anticoagulant.

Patient age according to the presence or absence of preoperative DVT was significantly different in THA patients ($P = 0.047$), but not in TKA patients (Table 3). In addition, preoperative walking ability in DVT+ THA patients was significantly lower than that in DVT− patients ($P = 0.038$). There were no statistically significant differences in sex ratio, BMI, and preoperative D-dimer values of both THA and TKA subjects (Table 3).

Incident DVT was observed in 211 patients (20.0%) at 1 week postoperatively among all patients; of these, 68 cases (6 at proximal and 62 at distal regions) were THA patients and 143 cases (16 at proximal and 127 at distal regions) were TKA patients (Fig 2). After 2 weeks of surgeries, 77 cases (7.3%) of newly-developed DVT were detected. Among them, 35 cases (6 at proximal and 29 at distal regions) and 42 cases (2 at proximal and 40 at distal regions) were seen in THA and TKA patients, respectively (Fig 2). There was no symptomatic PTE case in all patients postoperatively.

Fifteen cases of 39 preoperative DVT+ subjects developed postoperative DVT in THA patients, and the incidence rate was significantly higher compared with preoperative DVT

**Table 1. An image of the evaluations to the methods.**

**Three months before surgery: Investigation of preoperative DVT by lower extremity Doppler: Evaluation of VTE existence**

In the cases of large proximal thrombi, firstly, each patient consulted a certified specialist in the field, secondly, thoracic CT and CT angiography were performed to ascertain the existence of PTE, and finally, whether or not having surgery was possible had been evaluated by the certified specialists and our surgery team.

**One month before surgery: Re-investigation of preoperative DVT by lower extremity Doppler: Evaluation of VTE existence**

The patients with DVT 3 months ago had Doppler examination again to validate the thrombotic status.

**After surgery: Investigation of postoperative DVT by lower extremity Doppler: Evaluation of VTE existence**

A physical prophylaxis was conducted immediately after the surgery, and NOACs were administered from the day after surgery for 1 week. Lower extremity Doppler imaging was carried out at week(s) 1 and 2 postoperatively for all patients, and treatment was started upon the occurrence of DVT.

CT; computed tomography, DVT; deep vein thrombosis, PTE; pulmonary thromboembolism, VTE; venous thromboembolism

– subjects ($P$ = 0.0075) (Table 4). On the other hand, postoperative DVT occurrence was observed in 12 cases of 30 preoperative DVT+ TKA patients, but no statistically significant difference in the rate between the presence and absence of preoperative DVT was found ($P$ = 0.56) (Table 4).

## Discussion

In this study, we investigated the incidence of pre- and post-operative DVT in patients scheduled for lower extremity arthroplasty procedures and validated the effectiveness of our own treatment protocol for VTE. Preoperative DVT was detected in 6.5% of our cohort and those were located 1.4% at proximal and 5.1% at distal regions. In these preoperative DVT+ cases, NOACs were prescribed preoperatively and no detection of DVT was confirmed just prior to surgeries. Therefore, the procedures were safely performed even though the DVT had been found by Doppler ultrasound examination preoperatively. Patients with preoperative DVT planned for THA, were significantly older and had a lower walking ability. The incidence rate of DVT up to 2 weeks after surgery was 27.3%, with no occurrence of symptomatic PTE. A significantly higher rate of postoperative DVT development was observed in preoperative DVT + THA patients, but not in TKA patients only with a higher tendency.

VTE including asymptomatic and symptomatic DVT and/or PE have been considered as potentially serious complications after THA and TKA procedures. Since 75% of fatal PTEs are reported to be fatal within 1 hour of onset and the remaining 25% die within 48 hours of onset [7], preventing the development of DVT is highly important. On the other hand, the majority of VTEs are asymptomatic compared to symptomatic events, and that may be associated with long-term morbidity. Patients suffering from symptomatic VTE may have prolonged hospital

**Table 2. Prevalence rates of preoperative DVT.**

|  | Proximal | Distal | Total |
|---|---|---|---|
| THA, $n$ (501 patients 587 joints) | 9 (1.8%) | 30 (6.0%) | 39 (7.8%) |
| TKA, $n$ (553 patients 900 joints) | 6 (1.1%) | 24 (4.3%) | 30 (5.4%) |
| All cases, $n$ (1054 patients 1487 joints) | 15 (1.4%) | 54 (5.1%) | 69 (6.5%) |

DVT, deep vein thrombosis; THA, total hip arthroplasty; TKA, total knee arthroplasty

**Table 3. Characteristics of subjects according to the presence or absence of preoperative DVT.**

|  | Preoperative DVT | |
|---|---|---|
| THA | DVT (+) $n$ = 39 | DVT (−) $n$ = 462 |
| Age (years) | 75.9 | 72.3* |
| Sex (male/female) | 3/36 | 101/361[NS] |
| BMI (kg/m$^2$) | 23.5 | 23.8[NS] |
| Preoperative walking ability (JOA) | 5.42 | 7.13* |
| Preoperative D-dimer (µg/dL) | 1.34 | 1.27[NS] |
|  | Preoperative DVT | |
| TKA | DVT (+) $n$ = 30 | DVT (−) $n$ = 523 |
| Age (years) | 76.1 | 75.8[NS] |
| Sex (male/female) | 6/24 | 92/431[NS] |
| BMI (kg/m$^2$) | 23.9 | 24.1[NS] |
| Preoperative walking ability (KSS) | 50.0 | 49.4[NS] |
| Preoperative D-dimer (µg/dL) | 1.28 | 1.21[NS] |

DVT, deep vein thrombosis; THA, total hip arthroplasty; TKA, total knee arthroplasty; BMI, body mass index; JOA, Japanese Orthopaedic Association; KSS, Knee Society Score; NS, not significant

*$P$ < 0.05 (Student's $t$-test for age and Mann–Whitney U test for preoperative walking ability).

stay with a greater rate of re-hospitalization. Reduced quality of life and increased health-care expenditures will also occur as a consequence [8].

GCS and VF are generally used for intraoperative mechanical prophylaxis against venous thromboembolism (VTE) during total knee arthroplasty (TKA). There was a previous report that 47.3 % of TKA patients who underwent GCS and/or VF as the intraoperative mechanical prophylaxis developed deep vein thrombosis (DVT) in the non-operated extremities [9]. Fuchs et al. have reported that the incidence of DVT was significantly reduced from 25% to 3.6 % when an Arthroflow device (a passive ankle motion device) were used for prophylaxis against VTE [10]. Furthermore, Funayama et al. have described that the incidence of VTE after THA was 36.9 % in patients without any prophylactic measures, 15.6 % who underwent GCS and VF, and 1.0 % who received an intraoperative manual lower-leg massage and passive ankle motion [11]. In our study, GCS or VF based on our protocol was performed for 14 days or 3 days, respectively after surgery, and thereby causing 20% at 1 week and 7.3% at 2 weeks of DVT, which was not much different compared to the previous results [9–11]. These results suggest that own protocol for VTE prophylaxis would be useful to prevent VTE.

Fuji et al. have recently researched a Japanese healthcare database and reviewed that the incidence of DVT or PTE events was 1.3% or 0.2% for TKA and 0.9% or 0.2% for THA, respectively [5]. Meanwhile, Katsuki et al. reported that DVT prevalence rates in lower limb veins prior to arthroplasty were 11.6% at distal regions and 0.2% at proximal regions of 2878 Japanese patients [12]. In the present investigation, preoperative DVT was detected in 6.5% of subjects planned for lower extremity arthroplasties, and that was similar to previous studies [13, 14]; however, the rate at proximal regions (1.4%) was slightly higher than those previous reports. Preoperative DVT+ THA patients of our cohort were significantly older and less able to walk, and these trends have also been demonstrated in previous investigations [13, 14]. Hence, the elderly patients with diminished gait function should be paid much attention to VTE perioperatively.

According to the Japanese Orthopaedic Association guidelines, the incidence of post-THA VTE with the absence of prophylactic treatment will be 22–33% for DVT (9–17% for proximal

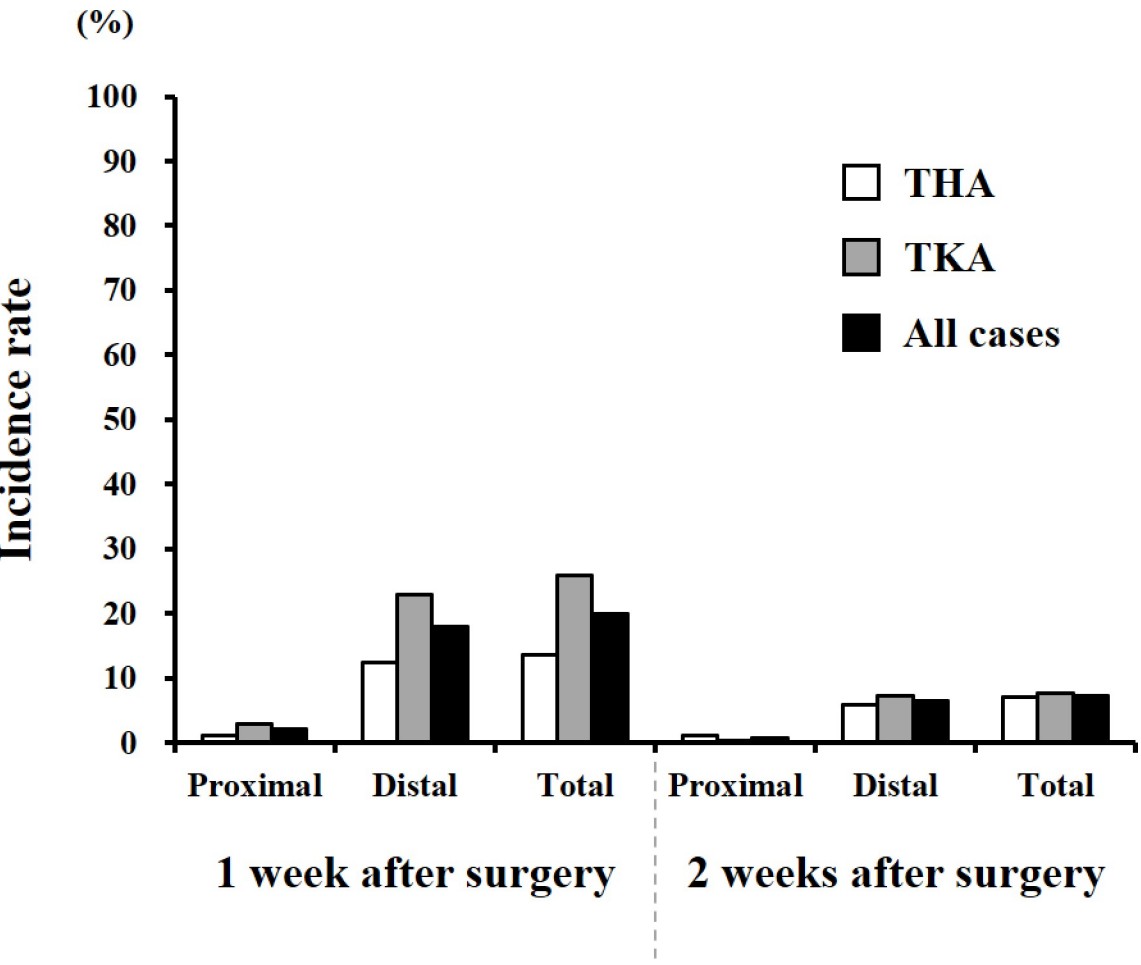

**Fig 2. Incidence rate of postoperative DVT.** Newly-developed DVT was examined at 1 and 2 week(s) after THA and TKA procedures. DVT, deep vein thrombosis; THA, total hip arthroplasty; TKA, total knee arthroplasty.

regions) and 0.9% for symptomatic PTE [15]. Considering that the rate of newly-developed postoperative DVT remained high (27.3%) in this study, our VTE prevention treatment protocol was not sufficiently effective in reducing DVT occurrence. Although physical prophylaxis and drug therapy were started on immediately after the surgery and the day after surgery,

**Table 4. Incidence of postoperative DVT according to the preoperative DVT status.**

| | Postoperative DVT | | |
|---|---|---|---|
| THA | DVT (+) $n$ = 103 | DVT (−) $n$ = 398 | *P*-value |
| Preoperative DVT (+) | 15 | 24 | < 0.01 |
| Preoperative DVT (−) | 88 | 374 | |
| | Postoperative DVT | | |
| TKA | DVT (+) $n$ = 185 | DVT (−) $n$ = 368 | *P*-value |
| Preoperative DVT (+) | 12 | 18 | 0.56 |
| Preoperative DVT (−) | 173 | 350 | |

DVT, deep vein thrombosis; THA, total hip arthroplasty; TKA, total knee arthroplasty
Significance of differences between DVT+ and DVT− groups preoperatively was evaluated by the Chi-squared test.

respectively, it was highly possible that DVT developed intraoperatively or within 24 hours postoperatively. Another reason for relatively higher incidence of DVT observed in the current investigation might be an aggregation of tiny clots up to 2 weeks after surgeries.

Note that the incidence of proximal DVT was 2.8% within 2 weeks postoperatively, which was relatively low, and appropriate treatment was immediately performed in each case. In addition, there was no case of symptomatic PTE, suggesting that our own preventive treatment protocol could be useful for reducing the occurrence of proximal DVT and fatal PTE.

In our cohort, plasma D-dimer levels of THA and TKA patients were not significantly different between DVT+ and DVT− groups preoperatively. D-dimer is a soluble fibrin degradation product that results from ordered breakdown of thrombi by the fibrinolytic system. The substance is generally employed as a marker of coagulation and fibrinolysis activation, and the assessment of circulating D-dimer has been shown to assist in the diagnosis of VTE [1]. Several studies have demonstrated that circulating D-dimer levels were elevated after TKA procedures for at least 10 days postoperatively, and there were no significant differences between the presence and absence of DVT [16, 17]. In recent study, Toner et al. reported that serum D-dimer remained high for at least 28 days and possibly considerably longer following TKA, and proposed not to use circulating D-dimer levels in patients with suspected VTE within this period [18]. Therefore, the regular monitoring by lower extremity Doppler images is considered very important for the early detection of postoperative DVT occurrence and the following immediate treatment initiation.

DVT is reported to occur in 1.4% while PE occurs in 1.1% of cases in Asia [19, 20]. Despite the relatively low incidence of DVT, this result is likely because ultrasound (US) was not routinely performed, only for symptomatic patients following major orthopedic surgery before hospital discharge [21]. In one routine US screening study, 26 patients (47%) had asymptomatic distal DVT after TKA [22], and 39 patients (9.6%) had asymptomatic DVT after THA [23]. In our study, our own protocol with US screening showed that the overall frequency of preoperative DVT was 6.5% (69 cases/1054 cases), but NOACs treatment caused no DVT prior to surgery. Also, postoperative VTE screening with US showed that DVT detection was 20% at 1 week and 7.3% at 2 weeks postoperatively. Note that there was no symptomatic PTE case in all patients postoperatively. Those findings suggest that VTE screening with US in our protocol could be useful to reduce postoperative DVT.

The strengths and limitations were as follows in this study: The strength was such large number of cases with 1,054 patients 1,487 joints. The limitations of this study were that 1) this study lacks appropriate controls and replication, 2) Since this preventive treatment protocol and investigation were performed at a single medical center, multiple-centered study will be needed for validating the effectiveness in arthroplasty patients.

## Conclusions

In the present study, the incidence of pre- and post-operative DVT in patients with lower extremity arthroplasty procedures were examined and the effectiveness of our own treatment protocol for VTE were validated. In the preoperative DVT positive cases, NOACs were prescribed and DVT was not detected prior to surgeries. VTE screening revealed that DVT at proximal regions was observed in 9 in 39 cases (23.1%) in THA and 6 in 30 cases (20%) in TKA preoperatively and 6 in 35 cases (17.1%) in THA and 2 in 42 (4.8%) in TKA 2 weeks after surgeries, showing that DVT development was suppressed after surgeries. Also, there was no occurrence of symptomatic PTE including fatal VTE. Those patients had been carefully observed with regular blood examination and continuous anticoagulant. Our findings suggest that our own VTE preventive treatment protocol could reduce the development of proximal

DVT, and the periodic Doppler examination as well as prompt treatment might prevent the fatal PTE.

## Acknowledgments

We would like to thank all participants in this study as well as Mr. Trevor Ralph for his English language editing. Our gratitude extends to Dr. Susumu Morioka and Dr. Hiroyuki Oshiba for their valuable advice regarding the present study.

## Author Contributions

**Conceptualization:** Takashige Momose, Yukio Nakamura, Masashi Nawata.

**Data curation:** Takashige Momose, Masaki Nakano, Takashi Maeda.

**Formal analysis:** Takashige Momose.

**Funding acquisition:** Yukio Nakamura.

**Investigation:** Takashige Momose.

**Methodology:** Takashige Momose.

**Project administration:** Yukio Nakamura.

**Supervision:** Yukio Nakamura, Masashi Nawata.

**Validation:** Masaki Nakano.

**Visualization:** Masaki Nakano.

**Writing – original draft:** Takashige Momose, Masaki Nakano, Yukio Nakamura.

**Writing – review & editing:** Takashige Momose, Masaki Nakano, Yukio Nakamura, Takashi Maeda, Masashi Nawata.

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
