## [Decision Letter · Decision Letter 0]

28 Sep 2023

PONE-D-23-23568Incidence and preventive treatment for deep vein thrombosis in total hip and knee arthroplastyPLOS ONE

Dear Dr. Nakamura,

Thank you for submitting your manuscript to PLOS ONE. After careful consideration, we feel that it has merit but does not fully meet PLOS ONE’s publication criteria as it currently stands. Therefore, we invite you to submit a revised version of the manuscript that addresses the points raised during the review process.

Please submit your revised manuscript by Nov 12 2023 11:59PM. If you will need more time than this to complete your revisions, please reply to this message or contact the journal office at plosone@plos.org. Please include the following items when submitting your revised manuscript:A rebuttal letter that responds to each point raised by the academic editor and reviewer(s). You should upload this letter as a separate file labeled 'Response to Reviewers'.A marked-up copy of your manuscript that highlights changes made to the original version. You should upload this as a separate file labeled 'Revised Manuscript with Track Changes'.An unmarked version of your revised paper without tracked changes. You should upload this as a separate file labeled 'Manuscript'.If applicable, we recommend that you deposit your laboratory protocols in protocols.io to enhance the reproducibility of your results. Protocols.io assigns your protocol its own identifier (DOI) so that it can be cited independently in the future. For instructions see: https://journals.plos.org/plosone/s/submission-guidelines#loc-laboratory-protocols. Additionally, PLOS ONE offers an option for publishing peer-reviewed Lab Protocol articles, which describe protocols hosted on protocols.io. Read more information on sharing protocols at https://plos.org/protocols?utm_medium=editorial-email&utm_source=authorletters&utm_campaign=protocols.

We look forward to receiving your revised manuscript.

Kind regards,

Eyüp Serhat Çalık

Academic Editor

PLOS ONE

Journal Requirements:

"NO"

"NO"

4. Please include your tables as part of your main manuscript and remove the individual files. Please note that supplementary tables (should remain/ be uploaded) as separate "supporting information" files.

Additional Editor Comments:

I congratulate the authors for this useful study. Your article is generally well designed and written. I have a small suggestion for your article: Add a paragraph at the end of the discussion section about the strengths and limitations of your work.

Your manuscript was reviewed by two reviewers, and their comments, which need to be addressed, are below.

Reviewers' comments:

Reviewer's Responses to Questions

**Comments to the Author**

1. Is the manuscript technically sound, and do the data support the conclusions?

Reviewer #1: Partly

Reviewer #2: Yes

2. Has the statistical analysis been performed appropriately and rigorously? 

Reviewer #1: Yes

Reviewer #2: Yes

3. Have the authors made all data underlying the findings in their manuscript fully available?

Reviewer #1: Yes

Reviewer #2: Yes

4. Is the manuscript presented in an intelligible fashion and written in standard English?

Reviewer #1: Yes

Reviewer #2: Yes

5. Review Comments to the Author

Reviewer #1: Your research is quite interesting because many patients with complaints of swollen legs are found due to prolonged static positioning before and after THA and TKA surgeries.

In the methods section, it would be advisable to provide clear information about what physical prophylaxis measures were undertaken on the patients and whether these measures were implemented only after the surgery.

In this study, did you not recommend the use of medical stockings? As we know, patients in a static or immobilized condition are often advised to use medical stockings. In this study, only medical stockings were used for 2 weeks after the surgery. Please provide further information regarding the protocol for using these stockings.

Reviewer #2: New method, well-founded study with methodological details adequately described.

Suggestions:

- Describe the sample calculation;

- Add an image of the evaluations to the methods;

- The study design was missing.

6. PLOS authors have the option to publish the peer review history of their article (what does this mean?). If published, this will include your full peer review and any attached files.

Reviewer #1: No

Reviewer #2: No

---

## [Author Response · Author response to Decision Letter 0]

13 Oct 2023

Dear Professor Eyüp Serhat Çalık

Academic Editor

PLOS ONE

 Thank you very much for your very constructive questions and suggestions.

We have substantially revised our manuscript according to the academic editor and the reviewers using tracking system as pointed out. 

We have slightly revised the manuscript title based on this revision. 

We now believe thar our manuscript be acceptable in PLOS ONE. 

Sincerely,

Yukio Nakamura, 

Journal Requirements:

→Thank you very much. We have confirmed the above. 

"NO"

→Thank you very much. We declare that “The authors received no specific funding for this work.”

"NO"

→Thank you very much. We declare that “The authors have declared that no competing interests exist.”

4. Please include your tables as part of your main manuscript and remove the individual files. Please note that supplementary tables (should remain/ be uploaded) as separate "supporting information" files.

→Thank you very much. We have revised this issue in the text. 

→Thank you very much. We have added new parts in the method and discussion sections with some new references raised by the reviewers. 

Additional Editor Comments:

I congratulate the authors for this useful study. Your article is generally well designed and written. I have a small suggestion for your article: Add a paragraph at the end of the discussion section about the strengths and limitations of your work.

→Thank you very much. We have added one paragraph at the end of the discussion　 about the strengths and limitations.　

Your manuscript was reviewed by two reviewers, and their comments, which need to be addressed, are below.

→Thank you very much. We have added the following in the discussion section: 

GCS and VF are used for intraoperative mechanical prophylaxis against venous thromboembolism (VTE) during total knee arthroplasty (TKA). There was a previous report that 47.3 % of TKA patients who underwent GCS and/or IPCD as the intraoperative mechanical prophylaxis developed deep vein thrombosis (DVT) in the non-operated extremities [Tateiwa T, Ishida T, Masaoka T, Shishido T, Takahashi Y, Onozuka A, Nishida J, Yamamoto K. Clinical course of asymptomatic deep vein thrombosis after total knee arthroplasty in Japanese patients. J Orthop Sug (Hong Kong) 2019;27(2):2309499019848095. ]. Furthermore, Funayama et al. have described that the incidence of VTE after THA was 36.9 % in patients without any prophylactic measures, 15.6 % who underwent GCS and IPCD, and 1.0 % who received an intraoperative manual lower-leg massage and passive ankle motion [Funayama A, Kitsuta Y, Fujie A, Tando T, Kanaji A, Toyama Y. Prevention of VTE in total hip arthroplasty with manual hand massage of the lower extremities after operation. Hip Joint. 2013;39:107–11.]. In our study, GCS or IPCD based on our protocol was performed for 14 days or 3 days, respectively after surgery, and thereby causing 20 % at 1 week and 7.3% at 2 weeks of DVT, which was not much different compared to the previous results. These results suggest that own protocol for VTE prophylaxis would be useful to prevent VTE. 

Reviewers' comments:

Reviewer's Responses to Questions

Comments to the Author

1. Is the manuscript technically sound, and do the data support the conclusions?

Reviewer #1: Partly

Reviewer #2: Yes

→Thank you very much for those critical but constructive suggestions. 

→Thank you very much for the wonderful comments and suggestions. As kindly suggested, it is best to describe technically solid scientific research with our data that support our conclusions. Actually, this is a retrospective study in which all of the patients performed with TKA or TKA were analyzed after surgeries with our own protocols (please see our figure and tables), thus, it is challenging to elaborate technically sound research data that support our conclusion. However, this is a first study with numerous patients’ data with own VTE preventive protocols which revealed relatively decreased DVT occurrence after surgery, although there have still been few reports on the preventive solid treatment protocol before and after arthroplasty procedures. Thus, we believe that our data will be useful for the readers who are experts in this fields in the PLOS ONE. 

We have added a limitation section which is that our study lacks appropriate controls and replication. 

We have revised the conclusion substantially as suggested. 

2. Has the statistical analysis been performed appropriately and rigorously? 

Reviewer #1: Yes

Reviewer #2: Yes

→Thank you very much. 

3. Have the authors made all data underlying the findings in their manuscript fully available?

Reviewer #1: Yes

Reviewer #2: Yes

→Thank you very much

4. Is the manuscript presented in an intelligible fashion and written in standard English?

Reviewer #1: Yes

Reviewer #2: Yes

→Thank you very much. 

5. Review Comments to the Author

Reviewer #1: Your research is quite interesting because many patients with complaints of swollen legs are found due to prolonged static positioning before and after THA and TKA surgeries.

In the methods section, it would be advisable to provide clear information about what physical prophylaxis measures were undertaken on the patients and whether these measures were implemented only after the surgery.

In this study, did you not recommend the use of medical stockings? As we know, patients in a static or immobilized condition are often advised to use medical stockings. In this study, only medical stockings were used for 2 weeks after the surgery. Please provide further information regarding the protocol for using these stockings.

→Thank you very much for those wonderful suggestions. We have substantially revised those suggested issues. In the method section, we have added the following: A venous foot pump (VF) (A-V Impulse™, Covidien, MN, USA) was applied from just after surgery to 3 days and gradual compression stocking (GCS) was used for 2 weeks after surgery. Mobilization was started on the next day of surgery, and then range of motion exercize and walking practice with physical therapists at the institutions were performed until discharge. Physiotherapy was not performed during this study. 

Reviewer #2: New method, well-founded study with methodological details adequately described.

→Thank you very much. 

Suggestions:

- Describe the sample calculation;

→Thank you very much for this helpful suggestion. Actually, compared to the previous review with 2 RCTs including 279 patients totally (Orthop Traumatol Surg Res. 2023 Apr;109(2):103364.), our study includes 1054 patients (202 males and 852 females) who underwent THA (501 cases, 587 joints) and TKA (553 cases, 900 joints), which is relatively high sample numbers. Even though our study is a retrospective one, we believe that our findings would be helpful for the readers.

- Add an image of the evaluations to the methods;

→We have added a methodological image as in the table 4. 

- The study design was missing.

→We have added the study design in the second paragraph in the method section. 

→Thank you very much. We have uploaded the figures using the PACE tool.

---

## [Editor Report · Decision Letter 1]

20 Oct 2023

Incidence and preventive treatment for deep vein thrombosis with our own preventive protocol in total hip and knee arthroplasty

PONE-D-23-23568R1

Dear Dr. Nakamura,

We’re pleased to inform you that your manuscript has been judged scientifically suitable for publication and will be formally accepted for publication once it meets all outstanding technical requirements.

Kind regards,

Eyüp Serhat Çalık

Academic Editor

PLOS ONE
---

## [Editor Report · Acceptance letter]

25 Oct 2023

PONE-D-23-23568R1 

Incidence and preventive treatment for deep vein thrombosis with our own preventive protocol in total hip and knee arthroplasty 

Dear Dr. Nakamura:

I'm pleased to inform you that your manuscript has been deemed suitable for publication in PLOS ONE. Congratulations! Your manuscript is now with our production department. 

Kind regards, 

on behalf of

Dr. Eyüp Serhat Çalık 

Academic Editor

PLOS ONE